# The impact of spaced learning within physics lessons in secondary schools

Yuxi Zhou[1], Rachel Hartley[2], Alessio Bernardelli[2], Andrew Tolmie[1]*

**1** Centre for Educational Neuroscience, Institute of Education, University College London, London, United Kingdom, **2** Institute of Physics, 37 Caledonian Road, London, United Kingdom

* andrew.tolmie@ucl.ac.uk

## Abstract

Spaced learning (SL) involves distributed learning with timed distraction breaks unrelated to the learning input. These breaks are thought to facilitate consolidation of novel information in long-term memory by accommodating post-activation neural recovery and biochemical changes in potentiation. Applications of SL have become commonplace across UK teaching and feature in government guidance, although there have been few research trials, especially within secondary schools, and none addressing physics independently. However, past research with 14–16-year-old students in England found that interspersing a fast-paced video input with ten-minute distraction breaks in a standard hour-long biology lesson yielded exam performance equivalent to regular teaching across many months. The present study adapted this regime to learning a novel physics topic, atomic structure, including radioactive decay, part of pre-16 compulsory science education in England. An in-class video covered all of the subject content in a single one-hour lesson, with three learning inputs, of length 11 - 14 minutes, interspersed with ten-minute distraction breaks. As well as 'business as usual' controls, the research included groups that only experienced the SL lesson, and those who were exposed to it prior to traditional teaching. Impact on learning was assessed using tests similar to state examinations taken at the end of a two-year physics course. Six non-selective state (non-fee charging) schools with varying demography were invited to join the study to ensure the sample was representative of the 93% of students who attend such schools in England. In total, 336 students completed all pre- and post-intervention (immediate and delayed) tests. Students studying combined sciences and separate physics were divided into groups cutting across conditions, with the former taking a shorter test. SL led to immediate benefits for separate physics students, but the SL only group showed no further gain at delayed post-test, with performance then equal to controls, as in past work. However, the SL plus group exhibited additional gains, 50% to 90% greater than the other groups at delayed post-test, following traditional teaching. Among combined science students, the SL plus group showed gains at delayed post-test 60% greater than controls. These effects were consistent across different schools. The implication is that SL provides a foundation for subsequent learning, often doubling its efficiency regardless of context, and that it does so with minimal additional input.

**Data availability statement:** All relevant data are within the manuscript and its Supporting Information files.

**Funding:** The author(s) received no specific funding for this work.

**Competing interests:** The authors have declared that no competing interests exist.

## Introduction

The application of cognitive neuroscience to school-based learning has grown considerably over recent years. Approaches including retrieval practice, interleaving and spaced learning – i.e., distributed learning opportunities coupled with distraction breaks – are now practiced across the teaching community in the UK and, indeed, feature in official government guidance and inspection criteria [1,2]. Many teachers have adopted some form of spaced learning technique in their classrooms, although typically in relatively loose fashion. Moreover, there have only been a limited number of research trials, especially within the context of secondary school teaching, and only one of these pertains to physics [3]. However, previous research by Kelley & Whatson [4] showed that for a GCSE biology module a concentrated hour-long lesson with the learning inputs interspersed with distraction breaks yielded the same performance in multiple-choice tests for an external exam as regular teaching across many months.

Kelley and Whatson's research provided the motivation for the present study, which explored the impact of similar spaced learning using video input within a single lesson on student knowledge of a novel topic, as compared to traditional teaching taking 10–12 hours. Between 2020 and 2022, the COVID pandemic necessitated a change in traditional teaching approaches and online video content became the norm in the UK, including within science education, where it was developed in part as an alternative to practical work in the laboratory. Building on this change in pedagogical approach, the present research was designed to measure the impact of video-based spaced learning within a standalone lesson in school, post-pandemic. A time-sequenced video input was created by Download Learning Ltd in partnership with the Institute of Physics (IOP) for use in a standard science lesson (see https://spark.iop.org/per-spaced-learning). We sought to ascertain whether a spaced-learning approach using video input for the knowledge-rich topic of atomic structure and nuclear decay improved a) learning outcomes in the absence of regular classroom teaching; and b) learning outcomes when used as a precursor to regular classroom teaching. An additional driver to this research related to teacher workload and well-being, given that workload is cited as an issue by 50% of science teachers planning to leave the profession [5].

### What does spaced learning involve?

Recent interest in the application of understanding from cognitive neuroscience within education is exemplified in the UK by the inclusion of aspects of such work in the Department for Education's Initial Teacher Training Core Content and Early Career Frameworks [1,2]. Within these, Strand 2 focuses on strategies for promoting good progress in learning on the part of students. Grasp of the role of working and long-term memory systems [cf. 6] – and the consolidation processes that underpin the shift from temporary to more permanent storage – is seen as central to such strategies, and the gateway to improving student learning and retention of curricular information. Key points emphasised in the Framework material are:

a)  regular and purposeful practice of what has previously been taught can help consolidate material and help students remember what they have learned

b)  requiring students to retrieve information from memory, and spacing practice so that they revisit ideas after a gap are also likely to strengthen recall.

The superiority of spaced learning compared to massed learning of large amounts of material has been researched and evidenced since the late 19th century [7]. However, understanding of why spaced learning can serve as a strategy to improve consolidation rests in particular on contemporary neuroscience research, as we detail below. This includes electrochemical and physiological work on memory systems within non-human species, especially rats, who

are long-established as providing a strong parallel to human brain function [8–10]. In what follows, we outline the main points regarding spacing effects that have been identified from such research.

It is long-established that traces of an event are initially stored in short term memory, but only for a limited period [11,12]. At a neural level, when a stimulus occurs, a neurotransmitter crosses the synapse between one neuron and the next, which causes channels in the destination neuron to open, allowing calcium ions in. This results in a positive charge which causes an action potential to fire, and the process is then repeated at the next neuron in line. When firing occurs, however, a biological chain reaction also follows, resulting in a transcription factor being manufactured and entering the nucleus of the neuron, creating mitochondrial or mRNA. This mRNA then travels out of the nucleus into the surrounding fluid, causing synapses that fired to strengthen, creating a short-term memory. The effects of this process, called synaptic tagging, are evident after ten minutes and last about three hours [13].

As the strengthening process takes time, when a second, similar stimulus occurs in quick succession, as in massed learning, it has not yet completed, and the second stimulus has no additional strengthening effect. However, if the second stimulus occurs after a time interval, the refractory (i.e., recovery) period, the synapses that were tagged the first time become further strengthened [7]. The more times the same neurons fire in this way, the stronger the synapses become [13]: repeated firing causes glia cells supporting the neuron to create myelin, a lipid insulation material, which allows the electrical signal to travel more quickly down the axon. The resulting Long Term Potentiation (LTP) at the dendritic spines of affected neurons creates more stable long-term memories [14]. Importantly, these potentiation effects extend beyond instances of exact stimulus repetition, which is rarely found in classroom environments. Lynch et al. [15] argue that the first presentation of a stimulus causes short term potentiation (STP) in some dendritic spines and primes (sensitises) others; when another similar stimulus is presented later, as in spaced learning, this not only further potentiates the original spines, but also those that were primed, allowing a degree of spread in both activation and potentiation.

Fields [13] found that time was the deciding factor in the shift from STP to LTP. Immediate repetitions of stimulation had no effect, and only when they were spaced out in 10-minute intervals was LTP created. EEG studies confirm activation patterns consistent with long-term retrieval following spaced repetitions [16,17]. The exact timing of repetitions is not entirely agreed upon, however, though the basic pattern is: Kramar et al. [7] found that the amount of potentiation was doubled when 60 minutes of delay was used compared to 10 or 30 minutes, but 90 minutes had no further effect. This echoes cognitive findings, where the accuracy of recalled items increases as the inter-stimulus interval (ISI) increases, reaches a maximum and then fades away [18].

In a review combining cognitive and molecular theories with computational modelling, Smolen et al. [17], point out that the majority of spacing studies have relied on a trial and error method to see which space is optimal. They considered whether it can be predicted instead from computer modelling, and cite research [19] that indicated irregularly spaced ISIs (in this instance using intervals of 10, 10, 5 and 30 minutes) produced the highest level of the proteins that induce LTP, and massed ISIs, the lowest. Modest irregularity might in fact fit reasonably well with the inevitable variations that occur in classroom implementations of spaced learning.

## Evidence of effectiveness in classroom-based research

Existing evidence on the impact of spaced learning within discrete lessons is limited. Kelley and Whatson [4] found that an effective translation of neuroscience evidence into classroom practice involved having three fast paced stimuli of around 12–15 minutes spaced by two

10-minute breaks. However, the limitations of their sample size (n = 67 in their experimental manipulation) and assessment format (solely multiple choice questions) preclude the drawing of firm conclusions. Similarly, a report by the Education Endowment Foundation (EEF) [20] compared three spaced learning approaches across the sciences. In **version 1**, day 1 involved three repeated 12-minute chemistry inputs with **10-minute spaces**; days 2 and 3 maintain the pattern, focusing on physics and biology respectively. **Version 2** had **24-hour spaces** between each subject input: there were three 12-minute chemistry, physics and biology inputs with no breaks between each input and a break at the end instead; these were repeated on days 2 and 3. **Version 3** had **10-minute spaces** between the chemistry, physics and biology inputs within the lesson **and 24-hour spaces** between each subject input. The research suggested that a combination of ten-minute and 24-hour spacing appeared to be the most promising but concluded that a larger trial was required to test this.

A subsequent large scale evaluation of these three approaches (referred to as SMART Spaces) was carried out by Hodgen et al. [21] in 125 state schools in England with over 14,000 students taking part in the trial. The study found that students in the spaced learning schools made no additional progress in attainment compared to those in the control schools. However, because of the large number of participating schools, the structure of the intervention was modified in a number of instances to fit in with schools' timetable and lesson structure. For example, some schools moved more quickly through the programme because their lessons were 50 minutes instead of a full hour in the design. Conversely, in schools with longer lessons teachers slowed down the delivery of the stimulus materials, by asking questions and expanding on points, thus extending 'active' periods beyond the intended 12 minutes. Both deviate from the planned spaced learning structure and potentially impacted negatively on students' encoding and retention of information, especially where the spaces between stimuli were shortened. Moreover, Hodgen et al. used spaced learning as a revision tool, rather than as an approach for learning novel content, and noted that many control group teachers reported using spaced learning practices as part of their initial teaching, creating a substantial confound.

At best then, prior work suggests that short, fast paced stimuli spaced by 10-minute breaks may facilitate encoding of new information in students' long-term memory [3,4], but this requires further systematic investigation in the context of novel learning, employing robust samples. Version 1 of the SMART Spaces design, which focuses on spacing content within the same curriculum area within a one-hour lesson also seems preferable, since it would generally provide the best fit to existing timetables.

## The present study

The research reported here therefore adopted Kelley and Whatson's spaced learning structure, using video learning inputs and timed 'distraction break' activities to ensure that the schools adhered to the precise timings required. Atomic structure was selected as the focal topic since this is a content-rich curriculum area, most of which is entirely novel for students aged 14–16 years. The intervention lesson consisted of three learning inputs, of length 11 - 14 minutes, interspersed by two 10-minute breaks where students did not engage with the learning content; it is crucial that the content of distraction breaks is completely unrelated to the SL content, since any further activation regarding the latter would mitigate directly against the recovery of potentiation on which SL rests. In a modification to Kelley and Whatson's design, as well as 'business as usual' controls, the research included groups that only experienced the spaced learning lesson, and those who were exposed to it prior to traditional teaching. Impact on learning was assessed using a set of multiple choice, short answer and longer answer test questions, in line with England and Wales General Certificate of Secondary Education (GCSE) examinations. Whether students were studying a combined sciences curriculum or taking

physics as a separate subject affected what test questions they could be expected to answer, so they were divided into two groups, cutting across conditions, with combined science students taking a shorter test.

The research addressed the question of how effective a one-hour spaced learning lesson was compared to an average of ten hours of traditional teaching for the atomic structure topic. It was hypothesised that:

1. The two intervention groups would show improvement in long term recall not exhibited by the controls.

2. The intervention groups would show immediate benefits within seven days of the spaced learning lesson.

We also addressed the question of what the learning gains were when the spaced learning lesson was used as a precursor to traditional teaching but formed no specific hypothesis relating to this in view of the absence of past research.

## Materials and methods

### Participants and trial design

Six UK non-selective state schools participated in the study (see Table 1 for the demographic characteristics of each), with recruitment starting on 9 July 2022 and ending on 2 September 2022. Two of these schools followed the combined GCSE science course and the remainder taught the GCSE triple award, with physics as a separate subject. The participating schools were recruited via direct invitation with a follow up webinar. In general, each provided three Year 10 classes (age 14–15 years): a Control Group (CG – traditional teaching), a Study Group 1 (SG1 – spaced learning only) and a Study Group 2 (SG2 – spaced learning plus subsequent traditional teaching). However, one school provided two classes in each condition, one school had insufficient numbers to provide a CG, and one school provided the target groups in Year 11 classes (age 15–16).

**Table 1. Demographic Indices for Participating Schools.**

| School | Prog 8[a] | Prog 8 LA[b] | % Grade 5 + English and Maths[c] | % Grade 5 + LA | FSM %[d] | Students on roll[e] | 6th Form[f] | IDACI Score[g] | Cohort (school year, GSCE science award |
|---|---|---|---|---|---|---|---|---|---|
| 1 | 0.08 | −0.12 | 65 | 47 | 10.6 | 1758 | YES | 0.05 | Year 10, Separate |
| 2 | −0.01 | 0 | 43 | 46 | 40.7 | 1588 | NO | 0.31 | Year 10, Combined |
| 3 | −0.24 | −0.14 | 40 | 48 | 24.6 | 1050 | YES | 0.34 | Year 10, Combined |
| 4 | −0.09 | 0.03 | 60 | 54 | 13.7 | 1141 | YES | 0.06 | Year 11, Separate |
| 5 | −0.38 | NA | 45 | NA | 15.6 | 1695 | YES | 0.13 | Year 10, Separate |
| 6 | 0.45 | 0.01 | 66 | 51 | 14.9 | 832 | NO | 0.05 | Year 10, Separate |

[a]Standardised index of how much progress students made across eight qualifications between the end of Key Stage 2 (11 years of age) and the end of Key Stage 4 (16 years of age) for the participating school.

[b]Average Progress 8 Score for state-funded schools in the same local authority, national average – 0.03.

[c]Percentage of students who achieved grade 5 or above in English and maths GCSEs for the participating schools.

[d]Percentage of students eligible for free school meals in the participating school, the average across all state schools is 23.8%.

[e]Number of students on roll at the participating school at the time of this research.

[f]Whether the participating school offers 6th Form education (age 16-18).

[g]Income Deprivation Affecting Children Index (IDACI) score for the Post Code of the participating school (https://www.find-npd-data.education.gov.uk/data_elements/00267ca3-f520-4cb0-a390-20a3bf575214)

Table 2 shows the structure of the trial. In total, 548 students were included, of whom a subset of 336 students completed all the appropriate pre- and post-intervention tests and were included in the final analysis. For students taking the combined science course, there were 136 students (40 in SG1, 54 in SG2 and 42 in the CG); and 200 students taking physics as a separate subject were included in the analysis (85 in SG1, 56 in SG2, and 59 in the CG). There were 235 participants in the experimental conditions, a substantial increase on the sample used by Kelley and Whatson [4]. Ethical approval for the research was provided by the Research Ethics Committee of the University College London Institute of Education (REC 1669). Written parental and student consent was obtained for all students who participated in the testing procedures.

## Materials

### Spaced learning lesson content

The video content for the lesson was designed by Download Learning Ltd (https://www.downloadlearning.co.uk/), who have significant experience with resource production for adult workplace-based training. The scripted content was developed in partnership with the IOP authors and reviewed by a team of active classroom teachers and expert physics teacher educators at the IOP (https://spark.iop.org/). The video consisted of three stimuli of between 11 minutes and 13 minutes and 30 seconds long, each with a different format and 'spaced' distraction activity included as part of a YouTube video. Whilst the stimuli had slight variations in timing the spaces were fixed at 10-minute intervals in line with the research noted above. The first stimulus used animated cards with a fast-paced narration to introduce atomic structure, radioactive decay, the different types of radiation and nuclear fission and fusion. In the second stimulus an expert physics teacher used a series of models and teacher exposition to revisit the themes introduced in stimulus 1. Stimulus 2 was filmed in a school teaching lab, with the relevant flash cards from stimulus 1 appearing on screen alongside the presentation. The final stimulus used a question and worked answer format; here, exam-style questions on atomic structure were posed to the students participating in the trial with a deliberately short response time, before presenting the step-by-step solution for each question. The purpose of this testing was for students to attempt retrieval of information from the previous sections of the video, rather than answer the questions in full. An outline of the sequence of the spaced learning video lesson is illustrated in Fig 1. The individual pictured in Fig 1 has provided written informed consent (as outlined in PLOS consent form) to publish their image alongside the manuscript.

Similar content was covered in each of the stimuli with each section providing a different context. In a carefully scaffolded sequence, the three sections addressed the whole of the Atomic Structure unit in the English GCSE physics specifications (equivalent to 10-12 hours of 'traditional teaching').

**Table 2. Trial Structure.**

| Study Group 1 | Study Group 2 | Control Group |
|---|---|---|
| All groups take pre-test prior to learning | | |
| 1 hour<br>Spaced Learning Lesson | | |
| 7-days after initial learning input<br>Both study groups take Test 1 | | |
| No teaching | Tradition Classroom Teaching | |
| 2 months after final learning input<br>All groups take Test 2 | | |

Between each of the atomic structure segments of the video, there were two recorded 10-minute distraction breaks unrelated to the stimulus content. It was impressed upon teachers at the recruitment stage and at the start of the trials that these breaks were essential to the spaced learning study and that the video should be viewed in its entirety to replicate conditions within each group of students and across all schools involved. Since the specific content of the distraction breaks is not important as long as it is distinct from the stimulus content, these breaks did not involve the whole-body movement activities employed by Kelley and Whatson's [4], in order to study the effects of spaced learning activities where it is not possible to engage in physical activity; to minimise disruption and potential behaviour management issues; and to allow the lesson to be developed, for example, for remote or independent use by students with reduced mobility. In the first distractor, students viewed a general knowledge quiz, with 'quick-fire' answers. For the second, students copied a complicated line drawing that was presented upside down. The pace and complexity of the tasks were designed to create sufficient cognitive demand to completely distract students from the intervening learning episodes.

## Tests

The questions used to assess students' prior and post intervention knowledge and understanding closely reflected the current style of physics exam questions in the UK GCSE exams. They included objective style multiple choice and short answer questions, in addition to one 6-mark extended response question, which requires a deeper level of understanding and reasoning. Kelley and Whatson's [4] test questions were all multiple-choice questions, which could be argued primarily tested recall rather than application of knowledge. The full test consisted of 13 items (see S1 File).

## Procedure

The resource was piloted in a state school setting to check the logistics of using an hour-long video input in a science classroom and the administration of the pre- and post-intervention tests. The pilot tests were marked by an external examiner and an IOP researcher

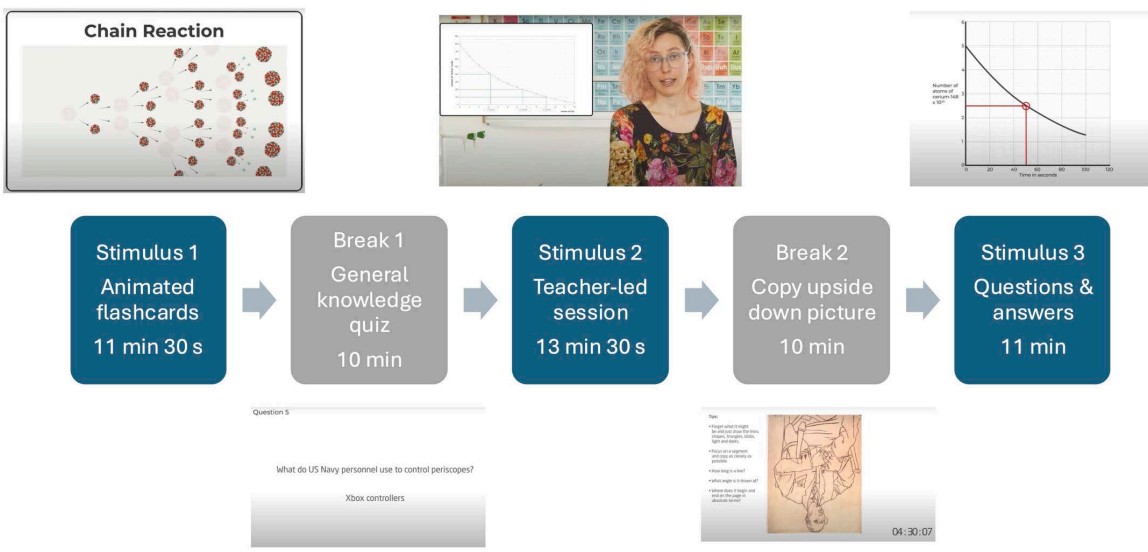

**Fig 1. An Outline of the Sequence of the Spaced Learning Video Lesson.**

independently to confirm the suitability of the mark scheme and to review the range of responses. All test papers within the subsequent trial were marked by the same external exam board consultant for standardisation purposes.

Following the pilot, the lead teacher from each participating school was invited to join an instruction webinar to explain the parameters of the research. Schools were sent paper copies for each set of tests, each with an anonymised student reference code and a spreadsheet to track student code allocations. An unlisted YouTube link to the spaced learning video was also shared with the lead teacher for use in the designated lesson. Email reminders with repeated instructions and clarifying updates were regularly sent to all participating schools to ensure teachers were running the programme as consistently as possible without direct supervision.

Each student completed the same test in class pre- and post-intervention, with the latter occurring two months after the final lesson of traditional teaching or the spaced learning, as relevant. Students in SG1 and SG2 completed an additional consolidation test with the same content seven days after the spaced learning lesson, i.e., in excess of the five days argued to be sufficient for initial encoding in long term memory [4,11]. Since traditional teaching typically contains retrieval opportunities and the CGs would therefore revisit the learning content in their regular lessons, we excluded the seven-day test for this group. Students following the separate physics route answered all 13 test questions in a 40-minute period, whilst students following the combined science route only answered the first 10 questions of the test and were allocated 30 minutes. The content of the SL lesson was the same for both routes, but contained some content extra to the combined science specification on which students on this route were not tested.

Tests were forwarded to an external marker via the IOP to preserve the anonymity of the trial schools. Question level data for each test was recorded against each unique student code in a password protected spreadsheet by the marker and this was stored in private Microsoft SharePoint folder with restricted access for subsequent data analysis.

## Analysis strategy

Items from the 10- and 13-item tests were first checked for reliability and coherence using Cronbach's alpha, and any ill-performing items were excluded to ensure the main analysis utilised robust indices.

Hypothesis 1, that the two intervention groups would show improvement not exhibited by the controls, was tested using a 3-way mixed ANOVA on pre-test and test 2 scores, with time as the within-subjects factor, group as the between-subjects factor of principal interest, and school as a further between-subjects factor to check whether any effects interacted with the student context. Tests were run separately for the students who had the combined science (basic) and separate physics (extended) item sets, given that these were not directly comparable.

Hypothesis 2, that the intervention groups would show immediate benefits from the spaced learning lesson, was tested using identically structured 3-way mixed ANOVAs on the pre-test and test 1 scores, using only the intervention groups. Results are reported below in this order, addressing outcomes for the basic item set students first, and then those for the extended item set students.

## Results

### Reliability tests

Cronbach's alpha was computed separately for items in the basic 10-item dataset and for those in the extended 13-item dataset. For the 10-item set, the alphas were an acceptable.705 for the

pre-test,.744 for test 1, and.795 for test 2. However, it was noted that with item 2 (a multiple-choice selection on the size of the atom written in standard form) excluded, the alphas improved substantially, to.744 for the pre-test,.973 for test 1 and.828 for test 2. For the 13-item data, the initial alphas were.609 for the pre-test,.804 for test 1 and.773 for test 2, but again the values improved when item 2 was excluded, to.627,.978 and.787 respectively. Overall, then, item 2 appeared to be something of an outlier that was not contributing to wider performance in the same way as other items, and it was decided to exclude scores on this for the main analyses and employ total scores across the remaining 9 (basic) and 12 (extended) items instead.

## Outcomes for basic (9-item) test

Table 3 shows the mean scores at pre-test, test 1 (where appropriate) and test 2 for the students who took the 9-item test, broken down by group. The 3-way mixed ANOVA on pre-test vs test 2 scores found significant main effects of group, $F(2,248) = 58.342$, $p < .001$, partial eta-square = 0.32, and of time, $F(1,248) = 30.390$, $p < .001$, partial eta-square = 0.11, but also a significant time x group interaction, $F(2,248) = 8.59$, $p < .001$, partial eta-square = 0.06, observed power = .99. However, in contrast to hypothesis 1, this reflected the fact that both the control group and SG2 showed a significant increase in scores from pre-test to test 2, $p = .004$ and $p < .001$ respectively, but SG1 did not. The controls performed significantly worse at pre-test than SG1 and SG2, $p < .001$ for both; their progress at test 2 might therefore be attributable to them simply making up ground to SG1 and SG2 pre-test levels. At test 2, SG2 performed substantially better than either the controls or SG1, $F(2,133) = 27.79$, $p < .001$, partial eta-square = 0.29, $p < .001$ for SG2 vs both other groups (see Fig 2 for a graphical presentation of the data).

There was also a significant main effect of school, $F(1,248) = 66.597$, $p < .001$, partial eta-square = 0.21, but no interaction between school and time, or between school and the time x group effect, indicating that context did not affect the overall pattern of progress, and that the general effects appeared to be reliably consistent.

Finally, the 3-way mixed ANOVA on pre-test vs test 1 scores found significant main effects of group, $F(1,168) = 19.684$, $p < .001$, partial eta-square = 0.10, and of school, $F(1,168) = 54.254$, $p < .001$, partial eta-square = 0.24, but no significant effect of time or interaction between school and time, i.e., there was no gain from pre-test to test 1 for either SG1 or SG2, regardless of context. For SG2, the bulk of their progress therefore took place between test 1 and test 2, after they received the traditional teaching.

## Outcomes for extended (12-item) test

The mean scores at pre-test, test 1 (as appropriate) and test 2 for the students who took the extended test are shown in Table 4. As for the 9-item test, the 3-way mixed ANOVA on pre-test vs test 2 scores found significant main effects of group, $F(2,382) = 6.313$, $p < .001$, partial

**Table 3. Mean Total Score (Out of 29) by Group on 9-Item Set at Pre-Test, Test 1 (Where Appropriate) and Test 2 (Including Change Values from Pre-Test to Test 1 and to Test 2).**

| | N | Pre-test mean(sd) | Test 1 mean(sd) | Change pre-test to test 1 | Test 2 mean(sd) | Change pre-test to test 2 |
|---|---|---|---|---|---|---|
| Control | 42 | 4.64 (2.63) | -- | -- | 7.50 (3.68) | 2.86 |
| SG1 | 40 | 8.05 (2.94) | 7.20 (3.74) | -0.85 | 7.88 (3.62) | -0.17 |
| SG2 | 54 | 9.41 (4.87) | 10.48 (4.85) | 1.07 | 14.02 (6.24) | 4.61 |

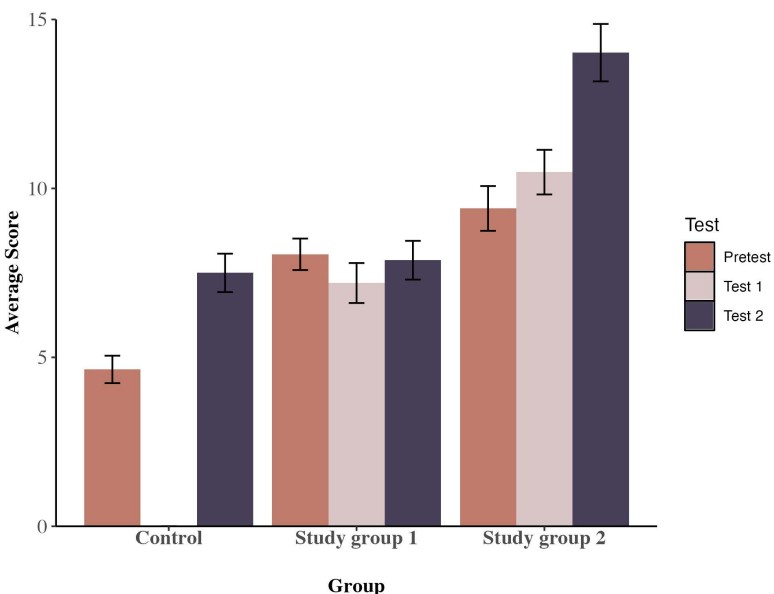

**Fig 2. Average scores for pre-test, test 1 and test 2 by group for the 9-item sample.**

**Table 4. Mean Total Score (Out of 39) by Group on 12-Item Set at Pre-Test, Test 1 (Where Appropriate) and Test 2 (Including Change Values from Pre-Test to Test 1 and to Test 2).**

| | N | Pre-test mean(sd) | Test 1 mean(sd) | Change pre-test to test 1 | Test 2 mean(sd) | Change pre-test to test 2 |
|---|---|---|---|---|---|---|
| Control | 59 | 9.75 (2.96) | -- | -- | 15.44 (6.05) | +5.69 |
| SG1 | 85 | 11.13 (4.03) | 14.56 (6.69) | +3.43 | 15.52 (6.69) | +4.39 |
| SG2 | 56 | 10.68 (4.60) | 13.86 (6.06) | +3.18 | 19.05 (6.79) | +8.37 |

eta-square = 0.03, and of time, F(1,382) = 140.990, p < .001, partial eta-square = 0.27, plus a significant time x group interaction, F(2,382) = 5.484, p = .004, partial eta-square = 0.03, observed power = .99. Unlike the 9-item results, though, there was no difference between any of the groups at pre-test; all the groups showed significant progress from pre-test to test 2, p < .001 in each instance; and at test 2, scores for SG2 were significantly higher than those for both other groups, F(2,197) = 6.01, p = .003, partial eta-square = 0.06, SG2 vs controls, p = .002, and SG2 vs SG1, p = .001, but those for the controls and SG1 did not differ (see Fig 3 for a graphical presentation). Overall, then, progress was substantially greater in SG2, consistent with the pattern in the nine-item test.

As before, there were significant effects of school, but these only interacted marginally with the condition x time effect, F (3,382) = 2.864, p = .037, partial eta-square = 0.02, indicating again that the general effects were reliably consistent.

The 3-way mixed ANOVA on pre-test vs test 1 scores revealed significant effects of time, F(1,268) = 34.616, p < .001, partial eta-square = 0.11, in contrast to the nine-item test, and no interaction with group, indicating commensurate effects in both. However, once again, for SG2, the bulk of their progress was test 1 to test 2, following the traditional teaching, with this combination producing the biggest gain, and one hour's spaced learning nearly doubling the

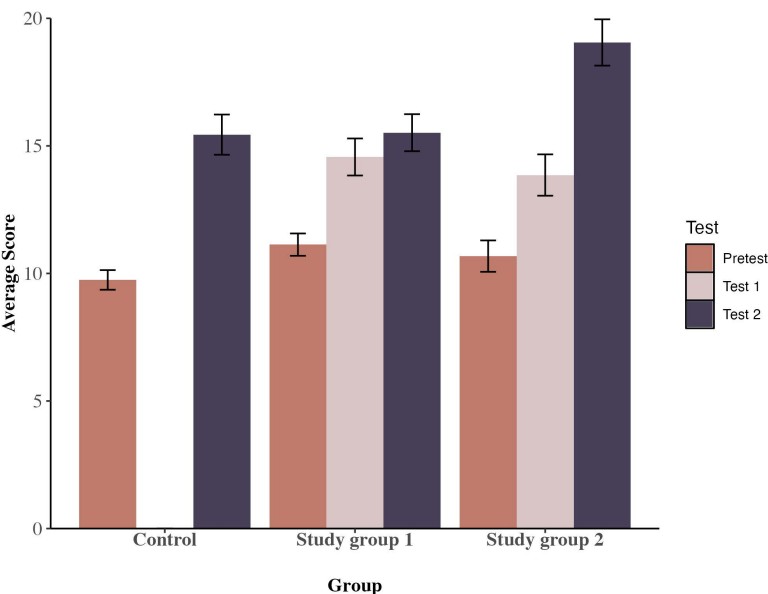

**Fig 3. Average scores for pre-test, test 1 and test 2 by group for the 12 items sample.**

impact of the traditional teaching experienced by the control group. There was a main effect of school, $F(3,268) = 19.100$, $p < .001$, partial eta-square $= 0.18$, and an interaction between school and time, but again this was only marginal, $F(3,268) = 2.795$, $p = .041$, partial eta-square $= 0.03$.

## Discussion

To summarise, SL led to benefits for the separate physics students taking the 12-item test seven days later, but for SG1 (SL only), there was little further gain at the test two months later, and performance then was equivalent to the controls (traditional teaching only). In contrast, SG2 (SL plus traditional teaching) showed additional gains at the two-month post-test, significantly outperforming the other two groups, with gains in test scores between 50% and 90% greater than that of the other two groups. For the combined science students taking the 9-item test, the pattern was slightly different, with SL producing no gain for either of the intervention groups at the seven-day post-test, with again no improvement at the two-month post-test for SG1. Once more, however, SG2 showed gains at the later post-test, following the added traditional teaching, with improvement some 60% greater than those who only had that teaching – though it is important to note that this group started from a lower baseline. Importantly, these effects were essentially consistent across different schools, though the effect sizes were also more robust for the combined science than the separate physics students.

The data suggest two key points. First, an hour's SL in physics on its own is capable of producing gains in understanding equivalent to a much longer period of traditional teaching, in line with Kelley and Whatson's results [4] – though apparently only for those students on the separate award physics programme, who are likely to have some greater baseline understanding. Those on the combined science route showed no ostensible benefit at either post-test, an outcome more in keeping with those found by Hodgen et al. in the SMART Spaces programme [21]. However, this pattern may be a function of the novel and relatively complex nature of the selected topic, atomic structure. This might have been harder to grasp in a short period for those on the combined science route, with less grounding; the pattern for a more

straightforward topic might have been more similar to that exhibited by the separate physics students.

Second, a brief episode of SL in this context is most effective not when used on its own, but when it is followed by more traditional teaching, producing a substantial boost in the impact of the latter. This pattern appeared to hold regardless of which route students were on. The most plausible explanation of this effect would appear to be that the SL period establishes a basic set of conceptual schemas for the topic area, with these being sufficiently explicit among more advanced students to produce immediate test gains but remaining more implicit and perhaps fragmented among those with less specialised understanding. The effect of subsequent teaching is to then consolidate and extend this understanding, building on the platform established by SL, and to render it more explicit for those with a lower baseline. These initial variations in grasp may explain, at least in part, the differences in observed effect size between the combined science and separate physics students; the gains were more robust for the former because the boost provided by SL was effectively more substantial and therefore more consistent.

This pattern of change would be consistent with Karmiloff-Smith's representational redescription (RR) model of conceptual growth [22], which frames progress in terms of a shift from initial implicit representations to more explicit, coordinated ones. The difference at the seven-day post-test for the combined and specialist students in the present study is further consistent with that reported by Philips and Tolmie [23]. They tested the RR model in the context of tutor effects on primary age children's understanding of the forces involved in balance, and also found more rapid progress in explicit understanding following initial input among those with more developed grasp of forces at baseline. The pattern observed in the present study is, of course, also consistent with neuroscience accounts of the effects of SL on consolidation and spread of activation [15,17]. At the same time, however, it suggests that past a certain point of long-term potentiation, subsequent activation over a much longer period with more distributed experience acts to increase this, in line with practice and retrieval accounts of consolidation [24,25]

The implication of the present data, then, is that SL serves to provide a foundation for subsequent learning, nearly doubling its efficiency; and that it does so with minimal additional input. Indeed, the hour taken by initial SL could readily be taken from a school's existing curriculum, increasing the impact of teaching at no extra cost (cf. [4] on the efficiencies presented by SL). What is less clear is whether further SL input on the same topic would produce further efficiencies or boosts to learning, with more pronounced effects. Although Hodgen et al.'s SMART Spaces programme [21] did involve repeated SL inputs, these addressed unrelated biology, chemistry and physics topics and focused on revision of material that had already been learnt. Even leaving aside the design weaknesses of this work noted earlier, it therefore is unable to provide an answer to this question.

More generally, further research is needed to examine: whether the effects of SL in combination with traditional teaching observed here extend to other topics; how far the impact of differences in baseline understanding on rate of progress indicated here shift according to degree of topic novelty and complexity; whether increasing the number of SL lessons on a given topic produces better or different patterns of outcomes; and how far effects vary with subject area and age group. We believe the present study serves to provide a strong impetus for these strands of research, however, by confirming and extending the work of Kelley and Whatson [4] on the benefits of SL within real classroom contexts.

## Supporting information

**S1 File. Test Items (Full Set).**
(PDF)

**S2 File. IOP_data_9items.**
(CSV)

**S3 File. IOP_data_12items.**
(CSV)

## Author contributions

**Conceptualization:** Rachel Hartley, Alessio Bernardelli.

**Data curation:** Yuxi Zhou, Rachel Hartley, Alessio Bernardelli.

**Formal analysis:** Yuxi Zhou, Andrew Tolmie.

**Investigation:** Rachel Hartley, Alessio Bernardelli.

**Methodology:** Yuxi Zhou, Rachel Hartley, Alessio Bernardelli, Andrew Tolmie.

**Project administration:** Rachel Hartley, Alessio Bernardelli.

**Supervision:** Rachel Hartley, Andrew Tolmie.

**Visualization:** Yuxi Zhou.

**Writing – original draft:** Yuxi Zhou, Rachel Hartley, Alessio Bernardelli, Andrew Tolmie.

**Writing – review & editing:** Yuxi Zhou, Rachel Hartley, Alessio Bernardelli, Andrew Tolmie.

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
