## [Decision Letter · Decision Letter 0]

14 Nov 2024

PONE-D-24-16225The impact of spaced learning within physics lessons in secondary schoolsPLOS ONE

Dear Dr. Tolmie,

Thank you for submitting your manuscript to PLOS ONE. As it is evident from the reviewers' reports, substantial revisions are needed before the paper can be considered for publication. One of the referees has raised significant concerns about the lack of novelty, alignment with global high school curricula, and depth of analysis, particularly regarding student backgrounds and comparative data from other regions. Additionally, the presentation of results would benefit from a clearer, more engaging graphical format. If you are confident you can address these concerns, we invite you to resubmit with detailed revisions and a response letter.

We look forward to receiving your revised manuscript.

Kind regards,

Mohammadreza Hadizadeh, Ph.D.

Academic Editor

PLOS ONE

Journal Requirements:

2. We note that your Data Availability Statement is currently as follows: “All relevant data are within the manuscript and in Supporting Information files.”

Please confirm at this time whether or not your submission contains all raw data required to replicate the results of your study. Authors must share the “minimal data set” for their submission. PLOS defines the minimal data set to consist of the data required to replicate all study findings reported in the article, as well as related metadata and methods (https://journals.plos.org/plosone/s/data-availability#loc-minimal-data-set-definition). For example, authors should submit the following data: - The values behind the means, standard deviations and other measures reported; - The values used to build graphs; - The points extracted from images for analysis. Authors do not need to submit their entire data set if only a portion of the data was used in the reported study. If your submission does not contain these data, please either upload them as Supporting Information files or deposit them to a stable, public repository and provide us with the relevant URLs, DOIs, or accession numbers. For a list of recommended repositories, please see https://journals.plos.org/plosone/s/recommended-repositories. If there are ethical or legal restrictions on sharing a de-identified data set, please explain them in detail (e.g., data contain potentially sensitive information, data are owned by a third-party organization, etc.) and who has imposed them (e.g., an ethics committee). Please also provide contact information for a data access committee, ethics committee, or other institutional body to which data requests may be sent. If data are owned by a third party, please indicate how others may request data access.

5. We are unable to open your Supporting Information file IOP_data_9items.sav and IOP_data_12items.sav. Please kindly revise as necessary and re-upload.

Reviewers' comments:

Reviewer's Responses to Questions

**Comments to the Author**

1. Is the manuscript technically sound, and do the data support the conclusions?

Reviewer #1: Yes

Reviewer #2: Partly

Reviewer #3: Yes

2. Has the statistical analysis been performed appropriately and rigorously? 

Reviewer #1: Yes

Reviewer #2: N/A

Reviewer #3: I Don't Know

3. Have the authors made all data underlying the findings in their manuscript fully available?

Reviewer #1: Yes

Reviewer #2: No

Reviewer #3: Yes

4. Is the manuscript presented in an intelligible fashion and written in standard English?

Reviewer #1: Yes

Reviewer #2: Yes

Reviewer #3: Yes

5. Review Comments to the Author

Reviewer #1: I found the article to be sound and well-expressed at all levels. Out of curiosity, would it be possible to publish direct links to the videos on atomic structure that were utilized? You mention the website for the company that produced the videos, but it's not clear that the actual teaching videos were available there.

Reviewer #2: As we can see in this article, the article is indeed written in an interesting way, but you can find definitely similar works in specific newspapers dealing with the teaching of science. The article is not innovative and does not reflect the field. The specific topic the author discussed in this manuscript does not reflect high school curricula in the world, and if it did, it should have been written and given a detailed explanation about it. In addition, it is not clear how this program is reflected in various other places.

Relatively speaking, the work is superficial and does not cover what is required of such work in this field. Some of the results are not displayed as desired, i.e. graphically. The results presented in the table should have been presented in a graph - something that was not done. The research lacks depth that teaches about the process and the background of the students. In addition, review groups from other places were missing.

The graphic presentation shown is very basic and it would be possible to present several graphs in a visual way that allows the overall picture to be provided.

Reviewer #3: lines 77-79: Include citations for the "contemporary neuroscience research". The authors also claim that rats are long established as "providing a strong parallel to human brain function." Include a citation for this claim, or if it is the authors' opinion, state it as such.

This is appropriate, especially in view of the fact that the authors do provide a citation for another "long-established" principle in lines 81-82.

The authors need to be more specific about what happens during a "distraction break". In lines 231-240, reference is made to a "quick-fire general knowledge test". Maybe include one of these in the supplementary materials. Also, more details are needed. What are the criteria for distraction break activities? Can it be anything, video games, etc." Is it necessary for the distraction break to have no relation at all to the course material?

The paper reads like a group of education experts (the authors) talking to another such group. It might be hard for an instructor with no prior experience to use this article to incorporate spaced learning into her classroom. To facilitate this, the authors may wish to explain terms such as in line 388: "immediate or delayed post-test", which are so obvious to them as to need no explanation. The comments above about being more specific about the distraction breaks also apply here.

If this reviewer understood Table 1 correctly, there will only be 12 minutes of material covered during an hour of contact time with the instructor, because the distraction breaks are followed with a repeated return to the material discussed during the first 12 minutes. In the US, a certain total amount of "covered" course material is mandated for each class. So if there is only going to be 12 minutes of new material introduced on each hour, it will be difficult to achieve the mandate. In this reviewer's opinion, there is already too much material required to be "covered" for each class, and this can only happen with a sacrifice of depth of learning in favor of (almost certainly temporary) breadth. Spaced learning seems to be a method for acquiring a greater depth of knowledge of a smaller amount of course material. Unfortunately, because of the way K-12 education is constructed in the US, spaced learning may not be feasible there. This reviewer regards that as unfortunate.

6. PLOS authors have the option to publish the peer review history of their article (what does this mean? ). If published, this will include your full peer review and any attached files.

**Do you want your identity to be public for this peer review?** For information about this choice, including consent withdrawal, please see our Privacy Policy .

Reviewer #1: **Yes: ** Jeremy C. Holtgrave

Reviewer #2: No

Reviewer #3: No

---

## [Author Response · Author response to Decision Letter 1]

8 Jan 2025

PONE-D-24-16225: The impact of spaced learning within physics lessons in secondary schools

Response to Reviewers

We thank the Editor and the three reviewers for their helpful comments and requests regarding our submission. We note below our response to each of the points raised and how these have been addressed within our revised manuscript and other elements of the resubmission. As requested, we have provided two versions of the revised manuscript, one with changes highlighted in yellow, the other without this markup.

Editor:

All files have been double-checked against the journal’s style requirements and corrected where necessary.

We note that your Data Availability Statement is currently as follows: “All relevant data are within the manuscript and in Supporting Information files.” Please confirm at this time whether or not your submission contains all raw data required to replicate the results of your study. Authors must share the “minimal data set” for their submission.

We confirm that all raw data required to replicate our results are provided within the manuscript and Supporting Information files, and that these meet the PLOS definition of the ‘minimal data set’ that we are required to share. The files IOP_data_9items.sav and IOP_data_12items.sav have been reformatted as .csv files in light of the Editor’s reported difficulty in opening the original versions, which should resolve that problem. Since the data are all provided as part of the submission, this obviates the need for contact information for a data access committee, ethics committee, or other institutional body to which data requests may be sent.

Please include your full ethics statement in the ‘Methods’ section of your manuscript file. In your statement, please include the full name of the IRB or ethics committee who approved or waived your study, as well as whether or not you obtained informed written or verbal consent. If consent was waived for your study, please include this information in your statement as well.

The full name of the Research Ethics Committee that approved the study is now included in the manuscript, along with the approval number (line 224). As stated before, written parental and student consent was obtained for all students who participated in the testing procedures; this text has been retained (lines 224-225).

Please include captions for your Supporting Information files at the end of your manuscript, and update any in-text citations to match accordingly. Please see our Supporting Information guidelines for more information: http://journals.plos.org/plosone/s/supporting-information.

Both the captions at the end of the manuscript and the in-text citations have been updated to the appropriate nomenclature.

We are unable to open your Supporting Information file IOP_data_9items.sav and IOP_data_12items.sav. Please kindly revise as necessary and re-upload.

We apologise for this problem. As noted above, the files have now been updated to .csv versions, which should resolve the issue.

We confirm that the files for each of our three figures have been checked using PACE, and that these checked versions are now the ones provided

Reviewer 1:

Would it be possible to publish direct links to the videos on atomic structure that were utilized?

We thank the reviewer for this suggestion and have now included a link in the manuscript (line 66) to the video which will be live imminently.

Reviewer 2:

You can find definitely similar works in specific newspapers dealing with the teaching of science. The article is not innovative and does not reflect the field.

We have rechecked existing literature in peer-reviewed journals (we assume the reviewer did not actually mean newspapers) and are still unable to find any work beyond that already cited which deals with evaluation of the systematic use of spaced learning to teach science in classroom settings.

We have revised the manuscript in various places (lines 23-26, 49, 57, 176-177, 258-263) to provide additional emphasis on the innovative aspects of our study:

• the use of spaced learning within a single lesson to cover a novel curriculum area

• the use of video to ensure rigorous stimulus content and timing relative to breaks in line with neuroscience research (see line 103)

• the application to physics

• the evaluation of the impact of this material both on its own and when deployed as a precursor to traditional teaching

To our knowledge, the first two aspects have only been addressed previously in the research we have cited (we have added further detail at lines 19-21, 52-53 and 143-149 – replacing Table 1 – to clarify the connections regarding this), and the remaining two aspects are unique, with the last being ground-breaking by going to the heart of the question of when use of spaced learning might be most effective.

The specific topic the author discussed in this manuscript does not reflect high school curricula in the world, and if it did, it should have been written and given a detailed explanation about it. In addition, it is not clear how this program is reflected in various other places.

We have provided additional detail on the focal topic of atomic structure and its relationship to the physics curriculum at lines 23-24, 28-29, 67-68 and 253-256. While we acknowledge that this topic derives specifically from the pre-16 compulsory science education curriculum in England, it is a crucial part of other high school physics curricula globally. As we also already acknowledged, though (lines 425-428, 466-468), it is important to examine the impact of spaced learning in other curriculum areas; although it is a rich topic and, in our view, provided an appropriate proving ground for the research we report, we make no claim for atomic structure being especially representative.

Relatively speaking, the work is superficial and does not cover what is required of such work in this field. Some of the results are not displayed as desired, i.e. graphically. The results presented in the table should have been presented in a graph - something that was not done. The graphic presentation shown is very basic and it would be possible to present several graphs in a visual way that allows the overall picture to be provided.

The key results were presented graphically as part of the original submission, in Figs 1 and 2 (now Figs 2 and 3), and we can only suppose the reviewer failed to notice these. As well as ensuring these figures adhere to PLOS standards, as noted above, we have now also included direct reference to them in the text at lines 355 and 383-384 in addition to the relevant captions, to ensure they are not overlooked.

The research lacks depth that teaches about the process and the background of the students.

Substantial detail about the background of the participating students was already provided as part of what is now Table 1 (Table 2 originally). We have left this intact but have added further detail in lines 29-31. We have added further detail on the process of the spaced learning lesson at lines 24-26, 176-177, 179, 236-249, and 258-263, and in a new Fig 1.

In addition, review groups from other places were missing.

We are unclear what the reviewer means here and are therefore unsure how to respond. If this is a reference to those who reviewed the curriculum-related content of the spaced learning video, we have added further detail on this at lines 234-235, but these necessarily had to be experts with respect to the physics curriculum in England as far as this particular exercise was concerned; reviewers from elsewhere could not be expected to be sufficiently versed in the details of that curriculum.

Reviewer 3:

Lines 77-79: Include citations for the "contemporary neuroscience research". The authors also claim that rats are long established as "providing a strong parallel to human brain function." Include a citation for this claim, or if it is the authors' opinion, state it as such. This is appropriate, especially in view of the fact that the authors do provide a citation for another "long-established" principle in lines 81-82.

The citations for the contemporary neuroscience research are the work that is described in what follows, as we have now made clear (line 89). We apologise for the lack of citations to support the use of rat models – this was an oversight, and we thank the reviewer for picking this up. A set of suitable citations have now been added (line 92 and lines 500-505).

The authors need to be more specific about what happens during a "distraction break". In lines 231-240, reference is made to a "quick-fire general knowledge test". Maybe include one of these in the supplementary materials. Also, more details are needed. What are the criteria for distraction break activities? Can it be anything, video games, etc." Is it necessary for the distraction break to have no relation at all to the course material?

Again, we thank the reviewer for prompting us to add helpful clarification. As the reviewer suggests, the key requirement is that the distraction activity has no connection to the material being taught. We have made this explicit at lines 180-183, 258-263 and in the new Fig 1.

The paper reads like a group of education experts (the authors) talking to another such group. It might be hard for an instructor with no prior experience to use this article to incorporate spaced learning into her classroom. To facilitate this, the authors may wish to explain terms such as in line 388: "immediate or delayed post-test", which are so obvious to them as to need no explanation. The comments above about being more specific about the distraction breaks also apply here.

We have now incorporated substantially more detail on the content and structure of the spaced learning lesson, as noted above, which we hope would help instructors to adopt systematic spaced learning approaches. We have also used more transparent language for some of the technical detail (lines 405, 407, 411, 412, 445). Some technical content is unavoidable (e.g. the data analysis procedures) in the interests of rigorous reporting, and difficult to explain to those unfamiliar with this without excessive length. However, descriptions of the research aimed at teacher audiences that adopt a more accessible format will also be made available in due course.

If this reviewer understood Table 1 correctly, there will only be 12 minutes of material covered during an hour of contact time with the instructor, because the distraction breaks are followed with a repeated return to the material discussed during the first 12 minutes. In the US, a certain total amount of "covered" course material is mandated for each class. So if there is only going to be 12 minutes of new material introduced on each hour, it will be difficult to achieve the mandate. In this reviewer's opinion, there is already too much material required to be "covered" for each class, and this can only happen with a sacrifice of depth of learning in favor of (almost certainly temporary) breadth. Spaced learning seems to be a method for acquiring a greater depth of knowledge of a smaller amount of course material. Unfortunately, because of the way K-12 education is constructed in the US, spaced learning may not be feasible there. This reviewer regards that as unfortunate.

We thank the reviewer for this interesting commentary. They are correct in thinking that the lesson essentially consisted of 12 minutes’ content, repeated three times in different ways, as the new Fig 1 makes clear. However, in terms of ‘amount’ of material covered, these 12-minute segments provided fast-paced delivery of the equivalent of ten hours traditional content (line 193), and as the results for Study Group 1 show, with equivalent effect, at least for the separate physics students. How this degree of coverage would be interpreted within US K-12 education is less clear to us, but we have already begun discussion with US colleagues on this and the points regarding curriculum equivalence and look forward to pursuing this discussion in more detail.

---

## [Decision Letter · Decision Letter 1]

9 Mar 2025

The impact of spaced learning within physics lessons in secondary schools

PONE-D-24-16225R1

Dear Dr. Tolmie,

We’re pleased to inform you that your manuscript has been judged scientifically suitable for publication and will be formally accepted for publication once it meets all outstanding technical requirements.

Kind regards,

Mohammadreza Hadizadeh, Ph.D.

Academic Editor

PLOS ONE

Additional Editor Comments (optional):

Reviewers' comments:

Reviewer's Responses to Questions

**Comments to the Author**

1. If the authors have adequately addressed your comments raised in a previous round of review and you feel that this manuscript is now acceptable for publication, you may indicate that here to bypass the “Comments to the Author” section, enter your conflict of interest statement in the “Confidential to Editor” section, and submit your "Accept" recommendation.

Reviewer #3: All comments have been addressed

2. Is the manuscript technically sound, and do the data support the conclusions?

Reviewer #3: Yes

3. Has the statistical analysis been performed appropriately and rigorously? 

Reviewer #3: I Don't Know

4. Have the authors made all data underlying the findings in their manuscript fully available?

Reviewer #3: Yes

5. Is the manuscript presented in an intelligible fashion and written in standard English?

Reviewer #3: Yes

6. Review Comments to the Author

Reviewer #3: Although I did not follow all of the technical details of the statistical analysis, as far as I could tell, it seems to be sound.

7. PLOS authors have the option to publish the peer review history of their article (what does this mean? ). If published, this will include your full peer review and any attached files.

**Do you want your identity to be public for this peer review?** For information about this choice, including consent withdrawal, please see our Privacy Policy .

Reviewer #3: No

---

## [Editor Report · Acceptance letter]

PONE-D-24-16225R1

PLOS ONE

Dear Dr. Tolmie,

I'm pleased to inform you that your manuscript has been deemed suitable for publication in PLOS ONE. Congratulations! Your manuscript is now being handed over to our production team.

Kind regards,

on behalf of

Dr. Mohammadreza Hadizadeh

Academic Editor

PLOS ONE